# Effects of Fluids on the Sublingual Microcirculation in Sepsis

**DOI:** 10.3390/jcm11247277

**Published:** 2022-12-08

**Authors:** Rachael Cusack, Susan O’Neill, Ignacio Martin-Loeches

**Affiliations:** 1School of Medicine, Trinity College, Pearse Street, D02 R590 Dublin, Ireland; 2Intensive Care Unit, St James’s Hospital, James’s Street, D08 NHY1 Dublin, Ireland

**Keywords:** sepsis, septic shock, microcirculation, resuscitation, fluid resuscitation, intensive care

## Abstract

Sepsis is one of the most common and deadly syndromes faced in Intensive Care settings globally. Recent advances in bedside imaging have defined the changes in the microcirculation in sepsis. One of the most advocated interventions for sepsis is fluid therapy. Whether or not fluid bolus affects the microcirculation in sepsis has not been fully addressed in the literature. This systematic review of the evidence aims to collate studies examining the microcirculatory outcomes after a fluid bolus in patients with sepsis. We will assimilate the evidence for using handheld intra vital microscopes to guide fluid resuscitation and the effect of fluid bolus on the sublingual microcirculation in patients with sepsis and septic shock. We conducted a systematic search of Embase, CENTRAL and Medline (PubMed) using combinations of the terms “microcirculation” AND “fluid” OR “fluid resuscitation” OR “fluid bolus” AND “sepsis” OR “septic shock”. We found 3376 potentially relevant studies. Fifteen studies published between 2007 and 2021 fulfilled eligibility criteria to be included in analysis. The total number of participants was 813; we included six randomized controlled trials and nine non-randomized, prospective observational studies. Ninety percent used Sidestream Dark Field microscopy to examine the microcirculation and 50% used Hydroxyethyl Starch as their resuscitation fluid. There were no clear effects of fluid on the microcirculation parameters. There was too much heterogeneity between studies and methodology to perform meta-analysis. Studies identified heterogeneity of affect in the sepsis population, which could mean that current clinical classifications were not able to identify different microcirculation characteristics. Use of microcirculation as a clinical endpoint in sepsis could help to define sepsis phenotypes. More research into the effects of different resuscitation fluids on the microcirculation is needed.

## 1. Introduction

Sepsis is a syndrome of pathological immune system derangement in response to an infectious agent, resulting in cardiovascular compromise and potential multi-organ failure [1]. The global incidence and prevalence of sepsis and septic shock is estimated to have a case fatality rate of 26–37% [2,3]. Guidelines and algorithms for treating sepsis have focused on early recognition, early empiric antimicrobial treatment and fluid resuscitation [4,5,6]. The microcirculation is significantly altered in sepsis, with disruptions in flow, reduced capillary density, microthrombi and increased heterogeneity [7,8]. The reported prevalence of microcirculation derangement in a heterogenous population in ICU is approximately 17% [9].

The microcirculation is the terminal effector site of the cardiovascular circulatory system, where tissue oxygen delivery and metabolic waste removal are coupled and controlled [10]. Together with its endothelium, it is also the largest organ system in the body, the average length of the capillaries in an average 70 kg man totaling ~6–15,000 km [10,11]. Terminal arterioles rather than capillaries themselves control the flow of blood and metabolic supply to organs. In sepsis there is mismatch of delivery and supply of oxygen to cells. Using Handheld Vital Microscopes (HVMs), we can see how the changes in the microcirculation such as stagnation, heterogeneity of flow, oedema and reduced haematocrit would affect the delivery of oxygenated haemoglobin to cells [8]. Microcirculatory oxygen delivery is not always improved by increasing mean arterial pressure (MAP), or oxygen saturation (SaO_2_) and this is haemodynamic incoherence [12]. Microcirculation disruption and failure is connected to morbidity and severity of organ dysfunction in conditions including haemorrhagic, cardiogenic and septic shock states [13,14,15]. Conversely, improvement in the microcirculation has been linked to improved outcomes in sepsis [16]. While more severe and persistent microcirculation derangement is linked to increased mortality in patients with sepsis, and the ability to resuscitate the microcirculation is connected to improved outcomes [15,17]. 

Improved techniques for examining the microcirculation and the ability to bring these techniques to the bedside in the intensive care unit (ICU) have enhanced our knowledge of the influence of sepsis on the microcirculation. The evolution of handheld vital microscopy to sublingual techniques makes it more accessible and applicable in clinical practice. Orthogonal polarisation spectroscopy has progressed to the more advanced sidestream dark field (SDF) and incident dark field (IDF) technologies [18,19]. SDF is currently the most popular and widespread method of microcirculation measurement at the bedside. It uses concentrically placed LEDs at a wavelength of 530 nm in a pulsed synchronous pattern to illuminate the microcirculation by haemoglobin absorption in red blood cells [18]. The benefit of IDF cameras is an increased field of view as well as improved visualisation of vessels to enhance the evaluation of proportion of perfused vessels and vessel density [19]. 

Despite over 20 years of research on the microcirculation we have not found a way to recruit it to improve oxygen delivery to cells in sepsis. A previous scoping review found that no vasoactive drugs evaluated had any effect on improving sublingual microcirculation parameters [20]. Fluid resuscitation improves survival in patients with sepsis [4,21,22]. While previous recommendations were to aggressively resuscitate hypotensive patients, new research into restrictive fluid strategies and de-resuscitation protocols are gaining prominence [23,24]. Strategies targeting the microcirculation in resuscitation are theoretically appealing but have yet to demonstrate benefit, indeed one study aimed at improving the microcirculation increased mortality of the intervention group [20,25]. In this systematic review we aim to assess the available evidence for the potential of fluid resuscitation to improve sublingual microcirculation parameters under direct visualisation.

## 2. Materials and Methods

### 2.1. Search Strategy

This review was conducted following a protocol established in advance published online at PROSPERO with registration number CRD42021285978. A systematic search of CENTRAL, EMBASE and PubMed/MEDLINE databases was undertaken from November 2021 to February 2022 by two independent data collectors. Various combinations of search terms “fluid resuscitation”, “sepsis”, “microcirculation”, “fluid therapy” and “sidestream dark field imaging” were carried out. We focused our review on techniques and devices used to look directly at the microvasculature and microvascular flow include orthogonal polarisation spectral (OPS) imaging/incident darkfield (IDF) microscopy/sidestream darkfield (SDF) microscopy. A search of ClinicalTrials.gov was also carried out to identify any relevant studies underway or completed but not published. Two authors undertook the search independently (RC and SON) who defined data sources, study selection, and data selection. Data was extracted from included reports using standardised data collection forms. After initial independent search and selection of papers both researchers met and discussed discrepancies in search results. Such discrepancies were resolved through discussion.

### 2.2. Selection Criteria

All papers found in the initial search were screened for relevance by title and abstract and compiled in Google Docs. Duplicates were excluded. We limited our search to English language papers or papers translated to English. Studies included focused on adult human subjects admitted to an ICU or Emergency Department with a diagnosis of sepsis or septic shock and requiring fluid resuscitation, as determined clinically. Interventions were any fluid bolus given to improve perfusion including crystalloids, colloids or blood products. Comparisons were made between the ability of diverse types of fluid or fluid bolus strategies to improve microcirculation parameters or measures of microvascular flow. Outcome measures were defined as an improvement in proportion of perfused small vessels (sPPV), perfused small vessel density (sPVD), total vessel density (TVD), proportion of perfused vessels (PPV), DeBacker density, microvascular flow index (MFI). We excluded review articles, conference abstracts and case studies (Figure 1, PRISMA flow).

### 2.3. Risk of Bias

Risk of bias was assessed according to the Cochrane guidelines on risk of bias assessment. RevMan version5.4 was used to collate studies. Standard methods described to assess RCTs were used to assess the quality of randomised studies. The Cochrane handbook describes and supplies templates for assessing bias in non-randomised studies [26]. The ROBINS-I tool was accessed, downloaded and completed for each study that did not fulfil randomisation. These results were then transcribed to RevMan to compare each study on standards applied to non-randomised study protocols(see Figure 2). The risk of bias tables can be viewed in the Appendix A.

## 3. Results

Our search of databases resulted in 3376 titles, which were copied into reference software manager and duplicates, papers that clearly bore no relevance to our research question and non-experimental review articles, or letters and editorials, were removed. This left forty-two unique reports for assessment to be included in the review. After reviewing the abstracts fifteen studies remained, (Figure 1). Reports were excluded because they were non-human animal studies, full reports with results were not available or not published, microcirculation variables were not reported, or they did not use handheld vital microscopy techniques. Table 1 summarizes the PICO characteristics and results of the included studies. 

### 3.1. Data

Six randomised controlled trials and nine prospective observational studies met the inclusion criteria.

### 3.2. Participants

The total number of participants in all trials was 813(mean = 50.6, min = 20, max = 207). Sepsis or septic shock = 14 studies (n = 763). One study investigated the effect of fluid bolus on ICU patients that were deemed clinically fluid responsive (n = 50). 

Studies included classified patients with sepsis according to Sepsis 2 or Sepsis 3 definitions [27,28]. 

Twelve studies were single centre studies and two studies included patients from two separate ICUs. One study was a multicentre trial across six centres. Two reports were post hoc analysis of microcirculation response in the same group of twenty patients in the same centre [29,30].

### 3.3. Intervention

One study compared 6% hydroxyethyl starch (HES) in 0.9% Sodium Chloride (NaCl) to 6% HES in 7.2% NaCl [31]. This study compared a 500 mL isotonic bolus to a 250 mL hypertonic bolus. Another study looking at hydroxyethyl starch compared HES 130/0.4 to normal saline 0.9% NaCl in an end goal directed therapy protocol [32]. Two studies used 500 mL 6% HES 130/0.4 or normal saline infused over thirty minutes, with a total n = 75 subjects [33,34]. One compared the microcirculation recruitment with fluid bolus to recruitment with passive leg raise manoeuvre [33]. Edul et al. examined the microcirculation in twenty-two patients who were given 10 mL/kg 6% HES in normal saline [35]. The largest trial by Massey et al. did not dictate what fluid to give and as a result 96% of patients enrolled received crystalloids, and 14.4% (EGDT), 8.3% (Protocol standard) and 7.5% (usual care) received RBC transfusion [36]. This was a sub-study of the ProCESS trial [37]. The analysis of patients does not distinguish between which fluid was received. The fluid volume was also not prescribed so at 6 h patients in the EGDT group received 2805 ± 1957 mL, patients in the Protocol standard group received 3285 ± 1743 mL and in the usual care group they received 2279 ± 1881 mL, *p* < 0.0001. By the 72 h time point, patients in the EGDT group received 7253 ± 4605 mL, patients in the Protocol standard group received 8193 ± 4989 mL and in the usual care group they received 6633 ± 4560 mL, *p* < 0.0001.

Four studies looked at the ability of red blood cells to recruit the microcirculation but compared microcirculation characteristics of groups of patients who responded, to those who did not [29,30,38,39]. One study compared the ability of Ringer’s lactate or 4% albumin to resuscitate the microcirculation in patients in the early (<24 h) or late(>48 h) phase of sepsis [40]. 

### 3.4. Comparator—Microcirculation Assessment and Outcome

We were particularly interested in how handheld vital microscopy technology is being transferred to the clinical environment. Of the fifteen studies included all but two utilised the Sidestream Dark Field Device (Microscan, Microvision Medical, Amsterdam, The Netherlands). This device is a descendant of the original Orthogonal Polarisation Spectroscopy machine and precedes the Incident dark field device, Cytoscan ARII (Cytometrics, Philadelphia, PA, USA), used by Sakr et al. and Van der Voort [39,41]. Two studies also examined peripheral thenar oxygenation with a NIRS device and one used temperature heterogeneity measurement (THI). 

The results of fluid bolus on the microcirculation were mixed, see Table 1. 

**Table 1 jcm-11-07277-t001:** Characteristics of included studies. TVD = Total vessel density, PVD = perfused vessel density, FCD = functional capillary density, sPPV = proportion of perfused small vessels, MFI = microvascular flow index, FHI = flow heterogeneity index, FC= fluid challenge, ICU = Intensive care unit, RBC = red blood cell, HES = Hydroxyethyl Starch, NaCl = Sodium Chloride/Saline, ↑ = improved/increased, ↓ = decreased/reduced.

Author	Centre	Participants	Fluid (Type)	Fluid (Volume)	Microcirculation Measurement Device	Microcirculation Outcome Measure
Dubin, 2010 [32]	2 Teaching ICUs Randomised controlled pilot study	Confirmed or suspected infection plus 2 ≤ signs of SIRS Signs of tissue hypoperfusion, MAP ≤ 65 mmHg despite fluid resuscitation or lactate ≤ 4 mmol/L n = 20	6% HES 130/0.4 in 0.9% NaCl vs. 0.9% NaCl	EGDT (CVP 8–12 mmHg, MAP ≥ 65 mmHg, ScVO2 ≥ 70%) Total fluid intake Saline group 8368 ± 2405 mL vs. 4682 ± 1371 mL *p* = 0.0008	Sidestream Dark Field Device (Microscan, Microvision Medical, Amsterdam, The Netherlands.)	At 24 h 6% HES 130/0.4 group; ↑ MFI ↑ PPV ↑ FCD ↑ TVD Saline group; ↑ Heterogeneity Index
Edul, 2014 [35]	Single centre, surgical ICU Prospective observational	Post-operative severe sepsis n = 22	6% HES in 0.9% NaCl	10 mL/kg	Sidestream Dark Field Device (Microscan, Microvision Medical, Amsterdam, The Netherlands.)	Before and 20 min after bolus ↑ RBC velocity Trend to ↑ PVD
Massey 2018 [36]	Multicentre, formal design sub-study of randomised control trial (ProCESS)	Adults with septic shock n = 207	Not assigned, 96% received crystalloid	EGDT n = 439 (2254 ± 1472 mL) vs. Protocol-standard n = 446 (2226 ± 1363 mL) vs. Usual care (2083 ± 1405 mL) *p* = 0.15	Sidestream Dark Field Device (Microscan, Microvision Medical, Amsterdam, The Netherlands.)	Reduced MFI in EGDT group, only stat significant difference Association of TVD, PVD and De Backer score with mortality
Ospina-Tascon, 2010 [40]	Single centre, ICU Prospective non-randomised observational	Severe sepsis, requiring fluid bolus n = 60	4% albumin (n = 31) vs. Ringer’s lactate (n = 29)	400 mL 4% albumin vs. 1000 mL Ringer’s lactate, over 30 min	Sidestream Dark Field Device (Microscan, Microvision Medical, Amsterdam, The Netherlands.)	↑ TVD ↑ PVD ↑ sPPV <24 h sepsis had more microvascular response to fluid
Pottecher, 2010 [33]	Medical and surgical University hospital ICU Prospective observational	Mechanically ventilated, severe sepsis or septic shock, requiring volume expansion n = 25	Crystalloid (0.9% NaCl, n = 8) or colloid (6% HES 130/0.4, n = 17)	500 mL over 30 min	Sidestream Dark Field Device (Microscan, Microvision Medical, Amsterdam, The Netherlands.)	PLR and volume expansion both resulted in the following microvascular changes; ↑ FCD ↑ MFI ↑ PPV ↓ FHI
Pranskunas, 2013 [34]	22 bed mixed ICU, prospective observational	Patients with clinical signs of impaired organ perfusion n = 50	0.9% NaCl (crystalloid) or 6% HES 130/0.4	500 mL/30 min	Sidestream Dark Field Device (Microscan, Microvision Medical, Amsterdam, The Netherlands.)	Low MFI vs. High MFI Low MFI group response to FC; ↑ MFI ↓ Clinical signs of hypoperfusion
Sadaka, 2011 [38]	54 bed medical surgical, university affiliated ICU Prospective observational	Severe sepsis RCC transfusion for Hb < 7.0 or Hb 7.0–9.0 with lactic acidosis or ScVO_2_ < 70% n = 21	Red Blood Cell (RBC)	1 unit	Sidestream Dark Field Device (Microscan, Microvision Medical, Amsterdam, Netherlands.) and Near Infrared Spectrometry (NIRS)(InSpectra Model 650; Hutchinson Technology Inc., Hutchinson, MN, USA)	No statistically significant change in PPV, MFI or PVD NIRS derived and SD derived variables changed in the same direction
Sakr, 2007 [39]	31 bed medical surgical ICU in a university hospital Prospective observational	Severe sepsis requiring RBC transfusion n = 35	RBC	1 unit n = 31 2 units n = 4 Groups divided based on > or <8% increase in capillary perfusion	Cytoscan ARII (Cytometrics, Philadelphia, PA, USA)	++Interindividual variability, overall, no impact on the microcirculation Improved microcirculation in patients with altered baseline, deterioration in patients with preserved baseline
Van Haren, 2012 [31]	15 bed ICU, Prospective, double -blind RCT	Septic shock n = 24	7.2% NaCl/6% HES (HT) vs. 6% HES (IT)	250 mL hypertonic vs. 500 mL IT	Sidestream Dark Field Device (Microscan, Microvision Medical, Amsterdam, The Netherlands.)	No significant changes found in any of the microcirculatory measurements compared to baseline or between groups ↑ Small vessel MFI in fluid responsive patients
Zhou, 2021, [42]	Emergency department and ICU Parallel group randomised prospective trial	Severe sepsis and septic shock n = 31	Not reported	POEM score guided vs. POEM score measured but did not guide resuscitation	Sidestream Dark Field Device (Microscan, Microvision Medical, Amsterdam, The Netherlands.)	Microcirculation guided resuscitation does not affect perfusion and organ function but does result in significantly lower fluid intake
Van der Voort, 2015 [41]	18 bed mixed medical surgical ICU, single-centre, open-labelled, randomised controlled pilot study	Severe sepsis and septic shock n = 90	Crystalloid and colloid (gelatin or albumin)	EGDT vs. resuscitation guided by microcirculation monitoring	Cytoscan (Cytometrics, Philadelphia, Pennsylvania, PA, USA)	No difference in SOFA between groups at Day 4
Donati, 2014 [29]	12 bed ICU, prospective randomised study	Sepsis, severe sepsis, septic shock requiring blood transfusion n = 20	RBC	Leukodepleted vs. non-leukodepleted RBC	Sidestream Dark Field Device (Microscan, Microvision Medical, Amsterdam, The Netherlands.), NIRS, PBR	PPV, DeBacker Score, MFI, HI Leukodepleted RCC = ↑ PPV, PVD, DeBacker Score&Blood velocity
Damiani, 2015 [30]	12 bed ICU, prospective randomised study	Sepsis, severe sepsis, septic shock requiring blood transfusion n = 20	RBC	Fresh leukodepleted vs. old leukodepleted vs. non-leukodepleted RBC	Sidestream Dark Field Device (Microscan, Microvision Medical, Amsterdam, The Netherlands.)	Change in fHB negatively correlated with TVD, DeBacker score, PVD change inverse correlation with fHB
Trzeciak, 2008 [16]	Single centre ED and ICU, prospective observational study	Septic patients treated with EGDT n = 33	Not reported	EGDT	Sidestream Dark Field Device (Microscan, Microvision Medical, Amsterdam, The Netherlands.)	MFI improved after EGDT in SOFA-improvers, but MFI did not improve in SOFA-non-improvers
Vellinga, 2013 [43]	Post hoc analysis of a single centre prospective observational study	Sepsis n = 70	Crystalloids, colloids, blood products	EGDT, 250 mL boluses, groups divided according to CVP > 12 mmHg vs. CVP < 12 mmHg	Sidestream Dark Field Device (Microscan, Microvision Medical, Amsterdam, The Netherlands.)	MFI increased after EGDT, MFI and PPV lower in CVP > 12 mmHg group

### 3.5. Outcomes—Randomised Trials of Fluid Resuscitation and the Microcirculation in Sepsis

Dubin et al. [32] carried out the first randomised trial comparing two different fluids’ ability to improve the microcirculation in patients with sepsis. They showed that after 24 h in the colloid group, who were randomised to receive 6% HES 130/0.4 had improved their MFI, PPV, FCD and TVD. The second randomised trial was by Van Haren et al. [31] found no significant changes in any of the microcirculation measurements compared to baseline or between groups. However, in fluid responsive patients the small vessel MFI did improve, indicating that fluid resuscitation can improve the microcirculation in a subset of patients. 

The third randomised study used simple randomisation of patients admitted to the ICU with sepsis or septic shock. Zhou et al. randomised patients to either receive microcirculation guided resuscitation or physician led resuscitation [42]. They used the Point-of-care Microcirculation (POEM) score, which is an ordinal scale from 1 (worst) to 5 (best). This score has shown reasonable reproducibility and interobserver reliability [44]. Doing the POEM score at the bedside negates the need for the assessor to leave and assess the microcirculation images elsewhere, increasing clinical utility. Assessors must be trained to use this composite assessment of flow and heterogeneity of four individual sublingual video-microscopy clips. They showed that POEM guided resuscitation did not affect organ function or urinary output, even though patients in the intervention group did receive less fluid than those in the physician guided group.

The Protocolized Care for Early Septic Shock (ProCESS) trial was published in 2014 and in 2018 the authors published a formal sub-study of the participants of the trial, looking specifically at microcirculation perfusion disturbances in septic shock [36,37]. Patients who had already been enrolled in ProCESS, a randomised trial of three different resuscitation strategies including blood, fluid, vasopressors and dobutamine. Patients were randomised according to the ProCESS protocol, the authors recruited patients with sepsis and signs of hypoperfusion. Using SDF imaging, the microcirculation was visualised, recorded and then assessed for effects of any of the randomly assigned treatment strategies. The authors enrolled 225 patients and analysed microcirculation data from 207. The three treatment groups were divided into EGDT, protocol based standard therapy, or usual care. The original ProCESS trial did not show an advantage of any resuscitation strategy over any other and similarly, this sub-study did not demonstrate any difference in the microcirculation. The investigators subsequently pooled the analysis of data to show that there was an association between poor microcirculation parameters and mortality (namely TVD, PVD, DeBacker density). These measures were lower at the 72 h time point in non-survivors. This is the largest trial examining microcirculation and in adjusted analysis found associations between increased CVP and higher TVD, De Backer score and HI, and an unexpected correlation between CVP and lower MFI as well as between MAP and lower TVD. 

### 3.6. Outcome Evidence from Prospective Observational Studies

Ospina-Tascon et al. used the SDF imaging device to show that patients in the first 24 h of sepsis admission had an increased microvascular response than patients more than 48 h into their septic episode [40]. This study had an n = 60 and compared 400 mL 4% albumin to 1000 mL Ringer’s lactate infused over 30 min. The online Appendix A showed that they found no statistically significant difference in microvascular response to either fluid. They also showed that after fluid resuscitation does improve microvascular markers and there was a significant difference from baseline in the sPPV, TVD, small vessel density, MFI and perfused small vessel density. 

Edul et al. gave post-operative severe sepsis patients 10 mL/kg of 6% HES in 0.9% NaCl (n = 22) [35]. They recorded images before, and 20 min following a fluid bolus and showed increased RBC velocity as well as a trend towards increased perfused vessel density. Interestingly, because they studied patients with a recent stoma formation, they demonstrated incoherence between the intestinal and sublingual microcirculation.

Pottecher et al. compared passive leg raise manoeuvre and volume expansion to show that the microcirculation reacted similarly to both interventions [33]. They showed a sublingual microcircualation improvement in FCD, MFI, PPV and reduced flow heterogeneity index. This was the only study that used an intrinsic volume expansion method such as the passive leg raise. The volume expansion component consisted of crystalloid versus colloid, however neither the main report nor the Appendix A online reports the results of a comparison of microcirculation to each of these products. 

Pranskunas et al. studied fifty patients with clinical signs of organ dysfunction and resuscitated them with either crystalloid or colloid [34]. In their analysis they grouped patients by those with a low MFI and those with a high MFI. They found that those patients with more deranged microcirculation parameters had a more significant response to fluid challenge (500 mL per 30 min).

### 3.7. Evidence for Red Blood Cells and the Microcirculation in Sepsis

Two included studies explored the reaction of the microcirculation to fluid bolus with red blood cells. Sakr et al. in 2007 looked at 35 patients admitted to ICU with sepsis or severe sepsis who required a blood transfusion according to criteria [39]. Thirty-one patients received a single unit of RBCs and 4 patients received two units. They reported significant inter-individual variability, with no significant overall impact on the microcirculation. Similarly, to Pranskunas et al., they commented that those patients with a more significantly deranged baseline microcirculation appearance improved more following transfusion. However, they also state that those patients who had a preserved microcirculation showed a deterioration after transfusion.

Sadaka et al. also recruited patients with sepsis or severe sepsis who required a blood transfusion because of low haemoglobin (<7.0 g/dL) or haemoglobin above 7.0 g/dL but accompanied by ScVO_2_ < 70% or lactic acidosis [38]. All patients in their study received only one unit of blood, however only eleven patients had their sublingual microcirculation assessed by the SDF camera. This group also reported no significant changes in sublingual microcirculation variables following RBC transfusion. Although they note that NIRS derived variables and SDF variables changed in the same direction.

Reports published by Donati et al. and Damiani et al. focus on the microcirculation effects of leukodepleted or non-leukodepleted RBCs in a group of 20 patients [29,30]. These reports are post hoc analyses of a group of patients recruited by Boerma et al. examining the effects of nitroglycerin on the sublingual microcirculation in sepsis [25]. The original cohort was randomised to receive nitroglycerin however these two studies observed the microcirculatory effects of the diverse types of blood products administered. Donati et al. found that blood flow velocity, MFI, PVD, DeBacker score and PPV increased in the patients who received leukodepleted blood, but that blood flow velocity and MFI decreased in the non-leukodepleted cohort. Damiani et al. further examined the effect of the age of the blood administered and found that increased levels of free haemoglobin (fHB) were negatively correlated with TVD, DeBacker score, heterogeneity index and PVD. Both of these studies are limited by their sample size and that they were not designed to look at this specific outcome in these patients. However, they provide interesting results about the potential negative effects on the microcirculation of non-leukodepleted RBCs and transfusion of older RBCs.

### 3.8. End goal Directed Therapy

Protocols for end goal directed therapy (EGDT) differ between centres. The studies included here that utilised EGDT for fluid resuscitation.

Trzeciak et al. published a prospective observational study in 2010 of the microcirculation in patients with sepsis who received EGDT in ICU [16]. This group analysed the sublingual microcirculation in patients at multiple timepoints as they were resuscitated. In thirty-three patients they found that MFI improved after EGDT in those patients whose SOFA improved, but EGDT did not improve MFI in SOFA non-improvers. The type of fluid and total fluid volume was not reported. 

Vellinga et al. found that even though the MFI increased in sepsis after EGDT, the MFI and PPV were lower in those patients who’s CVP was higher than 12 mmHg [43]. This study emphasises that the microcirculatory compartment is a low-pressure compartment that is affected by volume status and can be used to monitor volume resuscitation.

A study of ninety patients by Van der Voort et al. compared microcirculation guided resuscitation to EGDT in a single centre [41]. They found that there was no difference between SOFA scores at Day 4 after resuscitation. However, those patients resuscitated according to the microcirculation were more fluid positive and had longer ICU LOS. A major limitation of this study is that the groups were not matched for cause of sepsis and there was more abdominal sepsis in the microcirculation guided group, which introduces a confounding bias. 

### 3.9. Excluded Studies

Castro et al. published a randomised study in 2010 comparing resuscitation of septic patients guided by serum lactate or capillary refill time (CRT), peripheral surrogates of microcirculation measurement [45]. The authors standardised the measurement of CRT and also measured patients’ cardiac output with PiCCO or pulmonary artery catheter as well as examining SDF images of the microcirculation. The group showed no difference between groups according to fluid boluses, perfusion targets, 24 h SOFA scores or overall fluid balance. This study was not included because the authors did not compare the microcirculation before and after fluid resuscitation, except to report no difference in MFI between the groups at the 6 h mark. They concluded that CRT guided resuscitation is not superior to lactate targeted resuscitation based on fluid balance but that it had comparable effects on the microcirculation. 

An article published by Zhao et al. in 2012 investigated the clinical implications of dynamic monitoring of sublingual microcirculation changes in patients with severe sepsis [46]. This prospective study of 65 patients enrolled over six months in 2010 utilised SDF technology to measure TVD, perfused vessel density (PVD), PPV and MFI. The PPV and MFI were significantly reduced in the sepsis and severe sepsis patients but showed significant improvement immediately and 12 h after early goal directed fluid resuscitation. They used correlation analysis to show a significant negative relationship between PPV and prognosis. The authors concluded that SDF monitoring could be helpful to determine both disease severity and prognosis in sepsis and septic shock. They did report an improvement in the microcirculation following fluid resuscitation however only an abstract and not the full report of this study could be located. 

Lu et al. also studied microcirculatory effects of EGDT fluid resuscitation in sepsis but were only able to analyse images from 4 patients [47]. They found a trend towards increasing MFI and PPV in the patients after EGDT but there was no comparison group and a small sample size. 

Data from Veenstra et al. was also excluded as they did not provide information comparing the sublingual microcirculation in septic patients before and after fluid resuscitation [48].

## 4. Discussion

The aim of this review was to systematically answer the question does fluid resuscitation impact the sublingual microcirculation as measured by intravital microscopy in sepsis. The evidence we found consisted of heterogenous studies that could not be meta-analysed. These results fail to show a consistent, measurable improvement in the sublingual microcirculation following fluid resuscitation. 

### 4.1. Allocation (Selection Bias)

Four included studies used randomisation techniques to allocate patients to treatment groups. All four used concealed methods either web-based, computer based or opaque envelope techniques to ensure appropriate selection. The prospective observational studies included in this review did not have a way to reduce selection bias. The non-randomised studies included used a convenience sample or screened all admissions for inclusion, thereby recruiting patients in a certain timeframe. This method of selection is low in bias as all eligible patients would be included in the study.

### 4.2. Blinding (Performance Bias and Detection Bias)

Both the randomised and non-randomised studies allocated patients to receive different treatment strategies, therefore precluding blinding in this regard. Studies that allowed the treating physician to choose the fluid bolus therefore introduced bias in this way. Van Haren et al. blinded clinical staff to the selection of fluid delivered to patients. 

Assessment of microcirculation data was universally blinded. All included studies used a method of assessor blinding. The most common method was to code the video files of the microcirculation and present them to assessors non-sequentially so that no video could be associated as before or after an intervention and could not be associated with a particular patient or intervention arm. All studies reduced the effects of bias in outcome assessment.

### 4.3. Incomplete Outcome Data (Attrition Bias) and Selective Reporting (Reporting Bias)

All studies reported results from all patients and had complete data sets for all microcirculation outcomes. All studies were low in attrition and reporting bias. 

### 4.4. Other Potential Sources of Bias

As prospective observational studies that did not directly compare interventions there is potential for bias. Included studies operated a version of a before and after study where each patient acted as his or her own control. Studies included were from before and after the introduction of the Sepsis 3 definition, which may introduce bias in inclusion criteria and also limited our ability to compare and meta-analyse.

Sidestream dark field microscopy is a new field with guidelines for adequate image acquisition only recently developed and published. The expertise to operate and satisfactorily perform the measurements and analysis is limited and therefore there is a potential for bias, even though all authors report training outcome assessors appropriately. The method of obtaining microcirculation measurements has not been standardised. Most authors report obtaining 20 s videos from one sublingual area whereas others report obtaining three second video from five areas, or five sequences of 10 s and some authors did not report their acquisition protocol. Microcirculation measurement is at risk of multiple measurement bias where multiple readings could skew the result or preferential measurements could be reported. Microvision Medical, the developers of the Microscan SDF imaging device used in the majority of the included studies have proprietary software to aid in the objective assessment of microcirculation outcomes. Three included studies; Pranskunas et al., Pottecher et al. and Sadaka et al., utilised this AVA software to objectively assess the DeBacker density, PPV and PVD [33,34,38]. Edul et al. employed automated software techniques published elsewhere [35]. However, the majority of authors also reported MFI, which is a subjective measure that must be assessed by trained and experienced microcirculation experts. This is assessed by dividing the image of the microcirculation into quadrants and judging the flow through each visualised capillary and then giving the flow a score of 0 for no flow, to 3 for hyperdynamic flow. The MFI is then the average of the scores from each quadrant. Studies also reported the heterogeneity of flow index, which is also subjective. This score is usually calculated as the highest value for MFI minus the lowest MFI and divided by the mean MFI. Some studies used the average MFI or heterogeneity index of three to five videos as the reported value for a patient however some studies did not report how they chose what values to include in analysis.

All but two studies included in this review were published before the second consensus statement on the assessment of sublingual microcirculation in critically ill patients was released in 2018 [49]. This consensus statement includes recommendations on how to measure the microcirculation sublingually as well as how to interpret the images. These guidelines provide a necessary benchmark for future studies, unfortunately the advice and protocols could not be used to compare the studies included in this review. Using the consensus guideline, reproducible images should be achievable. The statement also recommends what measures should be reported, namely vessel density, perfusion indices and heterogeneity index [49]. Many of the studies reported here had different outcome measures reported, making comparison difficult. Interobserver reliability can also be an issue in many cases, as well as difficulties with consistent quality image acquisition [50,51]. With the introduction of the AVA software, used by some of the groups in this study, the assessment of images is now objective rather than subjective. Using the POEM score is another way that future studies can increase reliability, reproducibility and usability of the microcirculation at the bedside [44].

The ability to visualise the sublingual microcirculation is a great leap forward in point of care testing in ICU. Alterations in the microcirculation have been shown to be associated with poor outcomes and also to be prevalent in the ICU. A recent meta-analysis of the effects of therapeutics on the microcirculation found no particular benefit of any common ICU drugs [20]. This puts clinicians in a difficult position where they are able to visualise and measure an important patient characteristic, but we have not developed appropriate tools to treat the disturbance. Edul et al. published an interesting study that questions the coherence of different beds of the microcirculation [35]. This could mean that any treatments that do improve the sublingual circulation may not universally improve the circulation through vital organs such as the kidney, gut and brain. The sublingual region is easily accessible in the majority of patients with few contraindications to its use (e.g., post oral surgery, injury). However, because of the changes in the microcirculation between organs, concerns exist that what is visible with an SDF camera is not representative of the organs we are chiefly concerned with, namely the kidneys, liver, bowel and heart. There have been studies showing that the microcirculation of the postoperative septic bowel and sublingual region are not associated on day one but this relationship is re-established after 48 h [52]. This is possibly due to microcirculation shunting that occurs in sepsis [53].

The circulatory heterogeneity of the ICU patient population is a potential confounder preventing large ICU trials from demonstrating therapeutic effect. By delineating homogenous groups in ARDS, treatment effects could be isolated, and benefits shown, which is what is needed now to progress sepsis treatment [54]. Moving towards precision medicine means finding homogenous groups within large subsets of patients such as sepsis and septic shock. These syndromes have common elements but patients within may differ vastly, despite common pathophysiological features. One potential discriminating feature could be the microcirculation. Sakr et al. concluded that after analysis they had found no difference in the microcirculation after intervention but that there was a wide variety of interindividual responses [39]. This could of course mean that a subset of patients would have demonstrated a response if they had been placed in a subset for intervention. Pranskunas et al. separated their cohort into low and high MFI, showing that those in the low MFI group had a good response to fluid challenge [34]. Their work supports using the microcirculation to choose what patients receive a fluid challenge, aiming towards precision medicine.

The microcirculation changes over time in sepsis and resuscitation of macro-parameters may not be reflected in the delivery of oxygen at a cellular level [7,17]. The heterogenous results of the studies in this review indicate that the microcirculation behaves heterogeneously in critically ill and septic patients. Increasing the use of sublingual microcirculation monitoring to identify which patients will benefit from microcirculation targeted strategies is one possible avenue [55]. As Pranskunas et al. stratified patients according to pre-resuscitation MFI, this could be a novel way to distinguish patients in a precision medicine approach [34]. Microcirculation monitoring in sepsis is important because of the many pathways that can be deranged. Determining if hypoperfusion is due to reduced cardiac output or microcirculation stagnation will impact the potential treatments patients receive [56]. A recent publication highlighted the difficulty measuring the microcirculation in hypotensive patients. The authors reported a higher incidence of pressure artifact affecting their readings in hypotensive patients [57]. This raises the possibility of sicker, more shocked patients being assessed as having worse microcirculation outcomes. The most recent publication included, by Zhou et al., was the only group to report using the POEM score. The development of the POEM score encourages the use of the microcirculation without having to train clinicians in reading all the microcirculation variables. Their study highlights the feasibility of using the microcirculation to guide fluid resuscitation and supports a more conservative fluid therapy protocol [42]. The importance of de-resuscitation has been emphasised in recent years as morbidity and mortality have been shown to be associated with increased fluid intake in ICU [58]. This is true not only for sepsis but for heart failure, renal failure, respiratory failure or acute lung injury (ALI), sub-arachnoid haemorrhage and trauma and surgical patients [59,60,61]. Directing fluid resuscitation at the microcirculation could apply to several subsets of patients in the ICU population, more research in this area is needed.

The types of fluids examined in these studies has not sufficiently answered the question of colloids or crystalloids, which is best for resuscitation? Fifty percent of studies used Hydroxyethyl starch (HES) as their colloid resuscitation fluid and 1 used HES in a hypertonic solution. The European Medicines Agency (EMA) safety arm, Pharmacovigilence Risk Assessment Committee (PRAC) have suspended marketing authorisation for HES as of February 2022 [62]. This decision followed a series of reviews by the agency that restricted the use of HES to accredited institutions and appropriately trained specialists owing to the increased risk of kidney injury and death in certain patient populations in three commissioned reviews. In light of this controversy, HES is becoming rarer and rarer in clinical practice and therefore limits the applicability of these studies. Following the ALBIOS and SAFE trial, the surviving sepsis authors recommend including albumin as the resuscitative colloid of choice [63,64,65]. Ospina-Tascon et al. used 400 mL 4% albumin as one of their resuscitation fluids. In the online supplement the microcirculation comparison between albumin and crystalloid before and after bolus is shown to be non-significant [40]. However, Dubin et al. compared crystalloid and colloid effects on microcirculation and showed an improvement at 24 h, therefore further investigations into the efficacy of albumin in the microcirculation is warranted [32]. Albumin fluid resuscitation may benefit the glycocalyx, as shown in endotoxemic rats [66]. A recent review highlighted the importance of albumin as a carrier of sphingosine-1-phosphate and its effects on the endothelium, which may benefit the microcirculation of patients with sepsis [67]. There is a trial registered aiming to examine the effect of albumin, normal saline or HES on the sublingual microcirculation, trial registration NCT01319630 [68].

Sublingual measurement of the microcirculation may be confounded during fluid resuscitation by increasing right atrial pressure. Vellinga et al. undertook an interesting study that demonstrated that elevated central venous pressure is associated with impaired microcirculatory blood flow in sepsis [43]. Microcirculation perfusion pressure is post-erteriolar pressure minus venous pressure and becomes deranged in sepsis and septic shock. This could have implications for strategies that depend on higher CVP to indicate resuscitation end goals.

We chose to focus on the sublingual assessment of the microcirculation by HVMs. As reaching a macrohaemodynamic target does not ensure cellular oxygenation, intact capillary beds does not equate to oxygen delivery. Unless the microvascular architecture is intact you cannot be certain that you are adequately resuscitating the patient and shock may continue to worsen. Other clinical variables such as haematocrit, pO2 or SvO2 would be needed to calculate the microcirculatory oxygen delivery [69]. The microcirculation is highly variable and subject to change in multiple acute injuries, such as trauma and post-cardiac bypass [70,71]. Ensuring adequate microcirculation architecture is key to ensuring that other resuscitative efforts can focus on optimising cardiac output, oxygenation and haematocrit [10]. However, there are other bedside techniques to assess microcirculation integrity. Lactate is an important resuscitation target, indicating cellular dysoxia and metabolic mismatch. Unfortunately it is limited by confounding factors such as type a and b hyperlactemia, over production due to circulatory failure or reduced clearance [72]? It works well in conjunction with other markers of peripheral circulatory status such as capillary refill time and peripheral veno-arterial carbon dioxide gap to identify hyperlactemia as a result of hypoperfusion [73]. Acute kidney injury is associated with hypoperfusion states in sepsis and increased mortality [74]. Similarly, urine output and renal function can be a marker of microcirculatory derangement [75]. Other bedside markers of perfusion are of course important and can be used in conjunction with sublingual assessment. Capillary refill time has high interobserver reproducibility and responds to fluid resuscitation, making it a good candidate bedside resuscitation marker [76]. This is similar to the mottling score which is associated with 14-day mortality in sepsis [77]. The veno-arterial difference in partial pressure of carbon dioxide (Pv-aCO2 gap) is a good bedside marker of impaired tissue perfusion, however it does not differentiate between reduced cardiac output and microcirculation failure [78]. Although it is accessible, like the other markers it must be interpreted in the context of other bedside clinical indicators.

## 5. Conclusions

This systematic review of the literature has failed to identify good evidence that intravenous fluid can improve the sublingual microcirculation, measured using a HVM. Most of the studies included were done before the 2018 Consensus Statement on sublingual microcirculation was released, which will hopefully help to standardise studies in this area in the future. Fluid resuscitation is an important part of therapy for sepsis and septic shock, and the impact on the microcirculation must be considered. There is potential to use the sublingual microcirculation to stratify patients, prognosticate or to identify fluids and therapeutics that can repair the endothelial glycocalyx.

## Figures and Tables

**Figure 1 jcm-11-07277-f001:**
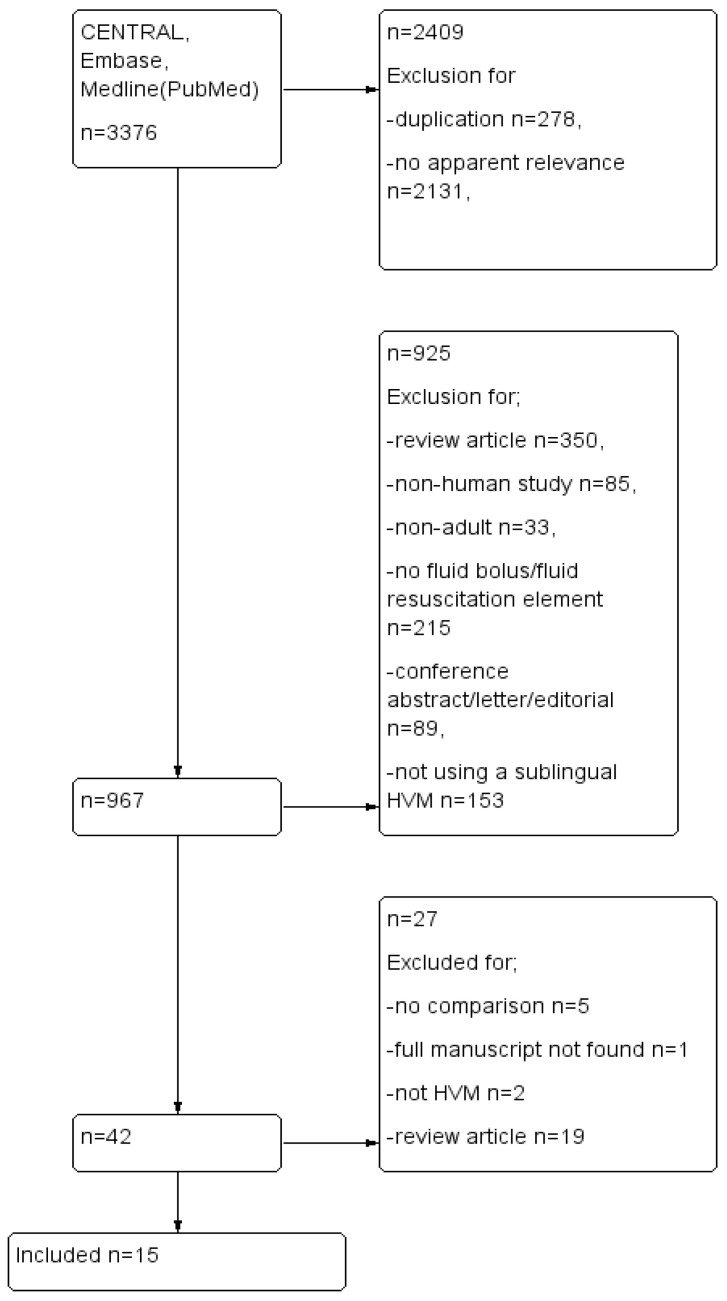
PRISMA flow diagram.

**Figure 2 jcm-11-07277-f002:**
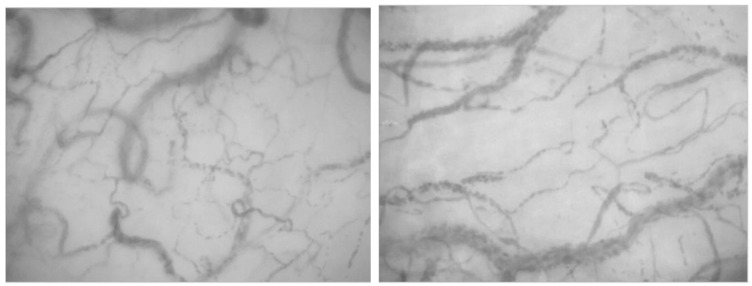
Sublingual microcirculation images from patients with sepsis, displaying characteristic heterogeneity of recruited vessels and reduced capillary density.

## Data Availability

Data available on request due to restrictions. The data presented in this study are available on request from the corresponding author. The data are not publicly available as they are kept on private Google docs file.

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
