# Peer review of "Effects of Fluids on the Sublingual Microcirculation in Sepsis"

_jcm, 2022, doi:10.3390/jcm11247277_

Round 1
Reviewer 1 Report
Introduction :
· Line 35: “The global incidence and prevalence of sepsis and septic shock is estimated to have a 35 case fatality rate of 26-30% [2]”.
More recent studies indicate an ICU mortality rate around 37%. And even found an ICU mortality around 50%, when focusing on study using the Third International Consensus Definitions for Sepsis and Septic Shock (Sepsis-3).
Vincent, J. L., Jones, G., David, S., Olariu, E., & Cadwell, K. K. (2019, May 31). Frequency and mortality of septic shock in Europe and North America: a systematic review and meta-analysis. Crit Care, 23(1), 196. https://doi.org/10.1186/s13054-019-2478-6
· Line 44: “Together with its endothelium, it is also the largest organ system in the body, 44 the average length of the capillaries in an average 70kg man totaling ~6-15,000km”
The reference does not seem to be the right one, there is no mention of this number in the reference 9.
· Authors should mention indirect assessment of the microcirculation:
A. It can be roughly done by arterial lactate level and its variations, however due to its well-known limitations, it has a poor correlation with microcirculatory disorders at the organ level.
Hernandez G, Bellomo R, Bakker J. The ten pitfalls of lactate clearance in sepsis. Intensive Care Med. 2019;45(1):82-5.
B. Urine output has also been considered a traditional marker of tissue perfusion (20) partially reflecting microcirculation. However, it may take time to assess and may be difficult to integrate.
Zafrani L, Ince C. Microcirculation in Acute and Chronic Kidney Diseases. Am J Kidney Dis. 2015;66(6):1083-94.
C. In particular, surrogate indirect microcirculation assessment that can also be done at the bedside using traditional markers of peripheral tissue perfusion signs such as capillary refill time (CRT), mottling, and ∆PCO2, are missing from this review.
Hariri G, Joffre J, Leblanc G, Bonsey M, Lavillegrand JR, Urbina T, et al. Narrative review: clinical assessment of peripheral tissue perfusion in septic shock. Ann Intensive Care. 2019;9(1):37.
Ltaief Z, Schneider AG, Liaudet L. Pathophysiology and clinical implications of the veno-arterial PCO2 gap. Crit Care. 2021;25(1):318.
à All these perfusion signs are strongly linked with microcirculatory blood flow alteration in septic shock.
· Line 71/72: “Despite over 20 years of research on the microcirculation we have not found a way to 71 recruit it to improve oxygen delivery to cells”.
This is true in septic shock, not really in cardiogenic shock.
den Uil CA, Caliskan K, Lagrand WK, van der Ent M, Jewbali LS, van Kuijk JP, et al. Dose-dependent benefit of nitroglycerin on microcirculation of patients with severe heart failure. Intensive Care Med. 2009;35(11):1893-9.
· Line 77 : Strategies targeting the microcirculation in resuscitation 77 are theoretically appealing but have yet to demonstrate benefit [19].
Authors should precise that it can even be dangerous sometimes in sepsis:
Boerma EC, Koopmans M, Konijn A, Kaiferova K, Bakker AJ, van Roon EN, Buter H, Bruins N, Egbers PH, Gerritsen RT, Koetsier PM, Kingma WP, Kuiper MA, Ince C. Effects of nitroglycerin on sublingual microcirculatory blood flow in patients with severe sepsis/septic shock after a strict resuscitation protocol: a double-blind randomized placebo controlled trial. Crit Care Med. 2010 Jan;38(1):93-100.
Materials and Methods
· Line 106/107 : “Studies included focused on human subjects admitted to an ICU or Emergency Department with a diagnosis of sepsis or 107 septic shock and requiring fluid resuscitation, as determined clinically ».
Authors should precise “on adult subjects”.
· Line 142/143: References should be given about the sepsis 2 and 3 definitions.
Results
· Line 149: “One study compared isotonic hydroxyethyl starch (6% HES in 0.9% Sodium Chloride NaCl) to hypertonic 6% HES in 7.2% NaCl[27]”.
Here and in other part of the text (such as line 156), this should be harmonized. Such has “X mL/kg of X% HES in X% NaCl”.
· Line 167: “Four studies looked at the ability of red blood cells to recruit the microcirculation 167 but compared microcirculation characteristics of groups of patients who responded, to 168 those who did not[31], [32]”.
The authors mention 4 studies, however, only there is only 2 references.
· Page 6 of 18, in Figure 1: the references don’t seem to be the right one just after the name of the first author. For example, “Dubin, 2010 [1]” while the number of this references is 28.
Besides, page 7 of 18, there is a little mistake with “Pottechar” instead of “Pottecher”.
· Line 185: “They showed in an n=20 that after 24 hours the colloid group, who were randomised to receive 6% HES 130/0.4 186 had improved their MFI, PPV, FCD and TVD” should be changed for “They showed that after 24 hours the colloid group, who were randomised to receive 6% HES 130/0.4 186 had improved their MFI, PPV, FCD and TVD”.
· Line 195: “They used the POEM 195 score, which is an ordinal scale from 1 (worst) to 5 (best).”
This score should be further developed, explaining the acronym and indicating the main reference.
· Line 201 to 219: References of the ProCESS trial and its ancillary study are missing here.
Discussion
· Authors should mention the latest consensus about microcirculation and why using it, it could help optimizing future study about microcirculation in sepsis:
Ince C, Boerma EC, Cecconi M, De Backer D, Shapiro NI, Duranteau J, Pinsky MR, Artigas A, Teboul JL, Reiss IKM, Aldecoa C, Hutchings SD, Donati A, Maggiorini M, Taccone FS, Hernandez G, Payen D, Tibboel D, Martin DS, Zarbock A, Monnet X, Dubin A, Bakker J, Vincent JL, Scheeren TWL; Cardiovascular Dynamics Section of the ESICM. Second consensus on the assessment of sublingual microcirculation in critically ill patients: results from a task force of the European Society of Intensive Care Medicine. Intensive Care Med. 2018 Mar;44(3):281-299.
· Among the recent study assessing surrogate indirect microcirculation parameters in sepsis, this study is of interest:
Raia, L., Gabarre, P., Bonny, V. et al. Kinetics of capillary refill time after fluid challenge. Ann. Intensive Care 12, 74 (2022).
· Other limitations regarding fluid therapy should be discussed, such as the increased right atrial pressure that is transmitted retrograde increasing venous pressure in vital organs. Indeed, mean capillary pressure appears to be more influenced by the downstream venous pressure than the upstream arterial pressure. In this perspective, central venous pressure appears to be one of the main determinants of capillary blood flow. This is of particular concern in some septic shock where the central venous pressure is often very elevated.
From this point of view, this article should be discussed:
Vellinga NA, Ince C, Boerma EC. Elevated central venous pressure is associated with impairment of microcirculatory blood flow in sepsis: a hypothesis generating post hoc analysis. BMC Anesthesiol 2013;13:17.
· A list of Abbreviations could be useful.
Author Response
Many thanks for taking time to review our manuscript and for your thoughtful comments. We have amended the manuscript and include our replies to your comments below.
Introduction :
- Line 35: “The global incidence and prevalence of sepsis and septic shock is estimated to have a 35 case fatality rate of 26-30% [2]”.
More recent studies indicate an ICU mortality rate around 37%. And even found an ICU mortality around 50%, when focusing on study using the Third International Consensus Definitions for Sepsis and Septic Shock (Sepsis-3).
Thank you for this suggestion, we will amend the manuscript to include it with the reference.
Vincent, J. L., Jones, G., David, S., Olariu, E., & Cadwell, K. K. (2019, May 31). Frequency and mortality of septic shock in Europe and North America: a systematic review and meta-analysis. Crit Care, 23(1), 196. https://doi.org/10.1186/s13054-019-2478-6
- Line 44: “Together with its endothelium, it is also the largest organ system in the body, 44 the average length of the capillaries in an average 70kg man totaling ~6-15,000km”
The reference does not seem to be the right one, there is no mention of this number in the reference 9.
Thank you, the second paper referenced in the bracket(reference 11) is the one that contains this figure:
Poole DC, Pittman RN, Musch TI, Østergaard L. August Krogh's theory of muscle microvascular control and oxygen delivery: a paradigm shift based on new data. J Physiol. 2020 Oct;598(20):4473-4507. doi: 10.1113/JP279223. Epub 2020 Sep 12. PMID: 32918749.
Authors should mention indirect assessment of the microcirculation:
- It can be roughly done by arterial lactate level and its variations, however due to its well-known limitations, it has a poor correlation with microcirculatory disorders at the organ level.
Hernandez G, Bellomo R, Bakker J. The ten pitfalls of lactate clearance in sepsis. Intensive Care Med. 2019;45(1):82-5.
- Urine output has also been considered a traditional marker of tissue perfusion (20) partially reflecting microcirculation. However, it may take time to assess and may be difficult to integrate.
Zafrani L, Ince C. Microcirculation in Acute and Chronic Kidney Diseases. Am J Kidney Dis. 2015;66(6):1083-94.
- In particular, surrogate indirect microcirculation assessment that can also be done at the bedside using traditional markers of peripheral tissue perfusion signs such as capillary refill time (CRT), mottling, and ∆PCO2, are missing from this review.
Hariri G, Joffre J, Leblanc G, Bonsey M, Lavillegrand JR, Urbina T, et al. Narrative review: clinical assessment of peripheral tissue perfusion in septic shock. Ann Intensive Care. 2019;9(1):37.
Ltaief Z, Schneider AG, Liaudet L. Pathophysiology and clinical implications of the veno-arterial PCO2 gap. Crit Care. 2021;25(1):318.
à All these perfusion signs are strongly linked with microcirculatory blood flow alteration in septic shock.
Thank you for this observation. We chose to focus on the sublingual microcirculation because of its potential future applications, as a lot of work has been done in this area recently. There are other reviews examining these other methods of bedside peripheral perfusion assessment and microcirculation. However, we will gladly add a section in the discussion regarding other methods of measuring microcirculatory reactivity and their uses in ICU.
- Line 71/72: “Despite over 20 years of research on the microcirculation we have not found a way to 71 recruit it to improve oxygen delivery to cells”.
This is true in septic shock, not really in cardiogenic shock.
den Uil CA, Caliskan K, Lagrand WK, van der Ent M, Jewbali LS, van Kuijk JP, et al. Dose-dependent benefit of nitroglycerin on microcirculation of patients with severe heart failure. Intensive Care Med. 2009;35(11):1893-9.
Thank you, we chose to focus this article on patients with sepsis and so are referring to this condition in this statement. We have rephrased for clarity.
- Line 77 : Strategies targeting the microcirculation in resuscitation 77 are theoretically appealing but have yet to demonstrate benefit [19].
Authors should precise that it can even be dangerous sometimes in sepsis:
Boerma EC, Koopmans M, Konijn A, Kaiferova K, Bakker AJ, van Roon EN, Buter H, Bruins N, Egbers PH, Gerritsen RT, Koetsier PM, Kingma WP, Kuiper MA, Ince C. Effects of nitroglycerin on sublingual microcirculatory blood flow in patients with severe sepsis/septic shock after a strict resuscitation protocol: a double-blind randomized placebo controlled trial. Crit Care Med. 2010 Jan;38(1):93-100.
Thank you for this suggestion, we have included a reference to this important study in the line 79.
Materials and Methods
- Line 106/107 : “Studies included focused on human subjects admitted to an ICU or Emergency Department with a diagnosis of sepsis or 107 septic shock and requiring fluid resuscitation, as determined clinically ».
Authors should precise “on adult subjects”.
Many thanks, amended
- Line 142/143: References should be given about the sepsis 2 and 3 definitions.
References to the 2001 and 2016 surviving sepsis meetings and publications added, many thanks.
Results
- Line 149: “One study compared isotonic hydroxyethyl starch (6% HES in 0.9% Sodium Chloride NaCl) to hypertonic 6% HES in 7.2% NaCl[27]”.
Here and in other part of the text (such as line 156), this should be harmonized. Such has “X mL/kg of X% HES in X% NaCl”.
Unfortunately, the authors of the study being referred to in line 149 did not use a ml/kg measurement that we can refer to as we have in line 156, however the volumes of fluid transfused have been inserted. We have done our best to harmonise the sentence and hope it has been improved by the rephrasing.
- Line 167: “Four studies looked at the ability of red blood cells to recruit the microcirculation 167 but compared microcirculation characteristics of groups of patients who responded, to 168 those who did not[31], [32]”.
The authors mention 4 studies, however, only there is only 2 references.
Thank you, this oversight has been amended and the references checked again.
- Page 6 of 18, in Figure 1: the references don’t seem to be the right one just after the name of the first author. For example, “Dubin, 2010 [1]” while the number of this references is 28.
Apologies and many thanks for this, the references have been checked and updated.
Besides, page 7 of 18, there is a little mistake with “Pottechar” instead of “Pottecher”.
The spelling has been corrected.
- Line 185: “They showed in an n=20 that after 24 hours the colloid group, who were randomised to receive 6% HES 130/0.4 186 had improved their MFI, PPV, FCD and TVD” should be changed for“They showed that after 24 hours the colloid group, who were randomised to receive 6% HES 130/0.4 186 had improved their MFI, PPV, FCD and TVD”.
Amended, with thanks.
- Line 195: “They used the POEM 195 score, which is an ordinal scale from 1 (worst) to 5 (best).”
This score should be further developed, explaining the acronym and indicating the main reference.
An explanation of the acronym, reference and the score has been added on line 200
- Line 201 to 219: References of the ProCESS trial and its ancillary study are missing here.
References added here, thanks.
Discussion
- Authors should mention the latest consensus about microcirculation and why using it, it could help optimizing future study about microcirculation in sepsis:
Ince C, Boerma EC, Cecconi M, De Backer D, Shapiro NI, Duranteau J, Pinsky MR, Artigas A, Teboul JL, Reiss IKM, Aldecoa C, Hutchings SD, Donati A, Maggiorini M, Taccone FS, Hernandez G, Payen D, Tibboel D, Martin DS, Zarbock A, Monnet X, Dubin A, Bakker J, Vincent JL, Scheeren TWL; Cardiovascular Dynamics Section of the ESICM. Second consensus on the assessment of sublingual microcirculation in critically ill patients: results from a task force of the European Society of Intensive Care Medicine. Intensive Care Med. 2018 Mar;44(3):281-299.
Thank you, a section discussing the impact of these guidelines has been added.
- Among the recent study assessing surrogate indirect microcirculation parameters in sepsis, this study is of interest:
Raia, L., Gabarre, P., Bonny, V. et al. Kinetics of capillary refill time after fluid challenge. Ann. Intensive Care 12, 74 (2022).
Thank you for signposting this very interesting paper, we have included measures of indirect microcirculation assessment in the discussion section.
- Other limitations regarding fluid therapy should be discussed, such as the increased right atrial pressure that is transmitted retrograde increasing venous pressure in vital organs. Indeed, mean capillary pressure appears to be more influenced by the downstream venous pressure than the upstream arterial pressure. In this perspective, central venous pressure appears to be one of the main determinants of capillary blood flow. This is of particular concern in some septic shock where the central venous pressure is often very elevated.
From this point of view, this article should be discussed:
Vellinga NA, Ince C, Boerma EC. Elevated central venous pressure is associated with impairment of microcirculatory blood flow in sepsis: a hypothesis generating post hoc analysis. BMC Anesthesiol 2013;13:17.
Thank you we have included a section on this viewpoint.
- A list of Abbreviations could be useful.
Thank you we will include a list of abbreviations for microcirculation indicators.
Reviewer 2 Report
The aim of this study was to assess available evidence for the potential of fluid resuscitation to improve sublingual microcirculation parameters under direct visualization. The evidence retrieved consisted of heterogenous studies that could not be meta-analyzed and the results failed to show a consistent, measurable improvement in the sublingual microcirculation following fluid resuscitation.
The topic is highly relevant form a clinical as well as a pathophysiological perspective. The methodology adheres to general principles and I think the report is worthy of publication but I do have some concerns.
The conclusion section is not consistent with the study objectives and main findings, but express the authors` general view. The scientific conclusion given at the beginning of the discussion is however, reasonable and well phrased. The present conclusion section must be edited.
The investigators have shown that the scientific evidence (for a role of sublingual microcirculation in the diagnosis, classification and monitoring of sepsis patients) is too weak and the study question is too limited to bear a big systematic review. I therefore suggest the following:
The authors should elaborate more on the evidence for a role of microcirculation assessment in sepsis. The weaknesses in the current assessment technologies should be addressed more broadly ie importance of the site of measurement; reproducibility of observations, etc; and does anatomical assessment of capillary beds provide all the information we need to estimate oxygen delivery to vital organs?
The paper has interest for a wide audience that is not familiar with technologies for assessment of microcirculation. The paper would therefore benefit from a photo of the capillary bed.
The above suggestions will require space and I therefore suggest moving figures about the methodology to supplementary materials.
Details: The meaning of the arrows in the table is not clear
Author Response
Many thanks for reviewing our manuscript, we have amended it as requested and include responses to your thoughtful comments below.
The aim of this study was to assess available evidence for the potential of fluid resuscitation to improve sublingual microcirculation parameters under direct visualization. The evidence retrieved consisted of heterogenous studies that could not be meta-analyzed and the results failed to show a consistent, measurable improvement in the sublingual microcirculation following fluid resuscitation.
The topic is highly relevant form a clinical as well as a pathophysiological perspective. The methodology adheres to general principles and I think the report is worthy of publication but I do have some concerns.
The conclusion section is not consistent with the study objectives and main findings, but express the authors` general view. The scientific conclusion given at the beginning of the discussion is however, reasonable and well phrased. The present conclusion section must be edited.
Thank you for taking the time to consider our review. We appreciate this comment and will amend the conclusions to be more appropriate.
The investigators have shown that the scientific evidence (for a role of sublingual microcirculation in the diagnosis, classification and monitoring of sepsis patients) is too weak and the study question is too limited to bear a big systematic review. I therefore suggest the following:
The authors should elaborate more on the evidence for a role of microcirculation assessment in sepsis. The weaknesses in the current assessment technologies should be addressed more broadly ie importance of the site of measurement; reproducibility of observations, etc; and does anatomical assessment of capillary beds provide all the information we need to estimate oxygen delivery to vital organs?
Many thanks for this suggestion, we have included a section on these points in the discussion.
The paper has interest for a wide audience that is not familiar with technologies for assessment of microcirculation. The paper would therefore benefit from a photo of the capillary bed.
Many thanks for this suggestion, we have moved the risk of bias tables to a supplementary materials section and have included a picture of the microcirculation from a SDF camera.
The above suggestions will require space and I therefore suggest moving figures about the methodology to supplementary materials.
Details: The meaning of the arrows in the table is not clear
Arrows have been added to the legend with the table.
Reviewer 3 Report
Dear Miss. Cusack,
here my comments:
Sepsis is new difined 2016, citations and explanations in the introduction should respect that new situation. The so called "sepsis-guidlines" fokus on microcirculation and two of them are from 2010, the most importend one, of the Surviving Sepsis Campaign (last one 11/2021), is not taken into account. The sepsis definition, also the septic shock one (now additional elavated lactate is necessary and not only vasopressor needed), changed 2016 (sepsis-3), so you have to take care if comparing studies vom bevore and after this timepoint.
The often used trem "the microcirculation" is to general for concrete statements, there are many different parameters that could be limited or improfed is a situation.
The reason of the microcirculatory limits, if the macrocirculation is adequarte, is not the volume, it is the septic process. The percentage of beside blood flow is importent, but additional fluid could just exess the beside flow. So also additional red blood cells have no efferct on the microcircular situation, the leukocytes in the PRB could influence the septic tissue.
First the macrocirculation must be intensively monitored and the cardiac index must be therapied in an adequarte range, if the microcirculation is not ok - this should be concuded. Then microcirculatr monitoring could help in fluid therapy and help to avoid edemas, whitch also limit microcirculation.
Generally the microcircularory monitoring helps to realise if the sepsis therapy is efficient, or not when the worse parameters do not improve over days. It helps not in guiding a special single medication.
Greetings
Author Response
Many thanks for taking time to review our manuscript, we appreciate your thoughtful comments, please find our replies below with recommendations included in the updated manuscript.
Sepsis is new difined 2016, citations and explanations in the introduction should respect that new situation. The so called "sepsis-guidlines" fokus on microcirculation and two of them are from 2010, the most importend one, of the Surviving Sepsis Campaign (last one 11/2021), is not taken into account. The sepsis definition, also the septic shock one (now additional elavated lactate is necessary and not only vasopressor needed), changed 2016 (sepsis-3), so you have to take care if comparing studies vom bevore and after this timepoint.
Thank you for this observation. We added a clarifying statement to the paper, we have referenced the 2 definitions and will add to the limitations that 2 studies are from after the Sepsis-3 definition publication on line 370.
The often used trem "the microcirculation" is to general for concrete statements, there are many different parameters that could be limited or improfed is a situation.
The reason of the microcirculatory limits, if the macrocirculation is adequarte, is not the volume, it is the septic process. The percentage of beside blood flow is importent, but additional fluid could just exess the beside flow. So also additional red blood cells have no efferct on the microcircular situation, the leukocytes in the PRB could influence the septic tissue.
We agree, the cause of microcirculation disturbance in sepsis is the septic process and perhaps even breakdown of the endothelium and microcirculation barriers themselves. Published works done on the effects of albumin on markers of endothelial damage and possible effects on restoring the endothelium and microcirculation integrity have been included following reviewer’s guidance at line 490. This article is warranted to show that there is not good evidence for colloids over crystalloids in restoring the microcirculation, but that work on albumin and better designed trials may be valuable.
First the macrocirculation must be intensively monitored and the cardiac index must be therapied in an adequarte range, if the microcirculation is not ok - this should be concuded. Then microcirculatr monitoring could help in fluid therapy and help to avoid edemas, whitch also limit microcirculation.
The reviewer is correct, and we have mentioned in the introduction that macrocirculation parameters might need to be complemented by a global approach including microcirculation.
Generally the microcircularory monitoring helps to realise if the sepsis therapy is efficient, or not when the worse parameters do not improve over days. It helps not in guiding a special single medication.
Thank you for your comments, the purpose of this review is to encourage further work in this area. I suppose it is hard to say if the microcirculation cannot be guided or helped by a single intervention if we do not have strong evidence for that. We have collected evidence that shows in some patients at certain times in their sepsis course, the microcirculation may be influenced by different fluid. Certainly the potential evidence that albumin could affect the microcirculation deserves to be investigated. We welcome your comments and highlight throughout the text as requested that research like our systematic review is necessary to advance the field in this area.
Greetings